# Development of a Novel Small-Scale Gust Generator Research Facility

**Zhenlong Wu [1,2,*], Tianyu Zhang [1], Yuan Gao [1] and Huijun Tan [1,*]**

1 College of Energy and Power Engineering, Nanjing University of Aeronautics and Astronautics, Nanjing 210016, China; tianyuzhang@nuaa.edu.cn (T.Z.); lupinthethirdred@163.com (Y.G.)
2 Integrated Energy Institute, Nanjing University of Aeronautics and Astronautics, Nanjing 210016, China
* Correspondence: zhenlongwu@nuaa.edu.cn (Z.W.); tanhuijun@nuaa.edu.cn (H.T.)

**Abstract:** In this paper, a novel small-scale gust generator research facility was designed and examined for generating Sears-type gusts. The design scheme, integration with the wind tunnel, experiment and validation of its capability are presented in detail. To help design the gust generator and validate the experimental results of the flow field characteristics generated by the developed gust generator, two numerical simulation methods, the field velocity method (FVM) and oscillating vane method (OVM), were utilized to detect the impacts of the geometrical parameters of the oscillating vanes and the downstream test model. The filtered experimental measurement results agree well with the numerical data, validating the capability of the developed gust generator to produce sinusoidal gusts. However, it should be noted that necessary measures are needed to prevent mechanical noise from interfering with the gusty flow field, which will be a focus of future research.

**Keywords:** gust generator; field velocity method; oscillating vane; Sears gust

## 1. Introduction

Gust is a prevalent and crucial atmospheric phenomenon in aviation, which has been a major factor in numerous catastrophic flight accidents over the past few decades [1]. The unsteady loads induced by gusts not only reduce the structural fatigue life of aircraft lifting surfaces, such as wings and horizontal tails, but also impact flight performance and passengers' comfort levels [2]. The presence of severe atmospheric gusts can have significant implications for the trajectory and propulsion system of an aircraft [3,4]. With the rapid advancement of computer technology, numerical approaches based on computational fluid dynamics (CFD) simulations have been widely applied in gust research, yielding numerous significant results that are challenging to attain through traditional experimental methods. Among the techniques employed for gust simulation, the oscillating vane method (OVM) and field velocity method (FVM) are extensively utilized. OVM produces gusts by simulating vane oscillation using a dynamic grid method. French et al. [5] used OVM to simulate gusts generated by a two-vane gust generator using the NACA0018 airfoil. Based on the simulation results, a two-vane gust generator is designed for use in a closed-return low-speed wind tunnel. Yigili et al. [6] used the OVM method to simulate the gust generated by a two-vane gust generator and compared the simulation results with those measured by Particle Image Velocimeter (PIV). The FVM method was initially proposed in 1992 by Singh et al. [7], who employed the unsteady Euler solution as a means to compute the indicial response of the wing when subjected to changes in angle of attack. Li et al. [8] applied the field velocity method in unsteady Reynolds averaged Navier–Stokes (URANS) simulations to predict the responses of an airfoil to arbitrarily shaped gust penetrations. Boulbrachene et al. [9]. derived the field velocity method and the velocity splitting method by applying the arbitrary Lagrange–Euler formula and the geometric conservation law in the context of a finite volume scheme. The differences between the two methods were analyzed and their advantages and disadvantages were compared. Li et al. [10] combined

the URANS solution, structural dynamics equations and the field velocity method to study the relieving ability of ordinary microjets on gust load. The impact of horizontal sinusoidal wind gusts on the flow field and aerodynamic performance of a serpentine inlet was investigated by Sun et al. [11] using the field velocity method, implemented in OpenFOAM. The significance of incorporating gust effect into the design and performance assessment of aircraft engine inlet was highlighted.

Gust loading is the basis of aircraft structural design or strength analysis. Experiments carried out in gust wind tunnels have made great contributions in solving aeroelastic problems [12], verifying numerical calculation methods [13] and exploring gust load alleviation (GLA) approaches [14]. A gust generator constitutes a crucial component of a gust wind tunnel, which has been extensively employed in the investigation of unsteady aerodynamic effects caused by atmospheric flows [15,16]. The utilization of gust generators is also prevalent in the experiment of gust responses of wings with high aspect ratios [17,18]. In addition, the topic of fluid–structure interaction (FSI) has been extensively studied by virtue of gust generators [19]. For low-speed wind tunnel experiments, in order to create a continuous gust environment and meet the experimental requirements of gust responses, the form of oscillating cascades is widely used, which are regularly driven by motors whose rotation of the output shaft is converted into the fixed-axis oscillation of the cascade by a multi-bar mechanism. Through a reasonable design of the transmission mechanism, the cascades can carry out pitching oscillations with expected frequencies and amplitudes. Buell [20] conducted an experimental study on the vane arrangement in a cascaded gust generator. His findings indicated that, when each cascade moved in phase, the pulsation velocity primarily occurred on the side of the wind tunnel centerline, and the amplitude of vane oscillation was directly proportional to the gust amplitude. Sergio [21] designed a vane gust generator to experimentally validate the effectiveness of an aircraft wing control system in suppressing flutter and mitigating gust loads. Liu et al. [22] used nonlinear elastic response analysis combined with gust wind tunnel experiments to study the aeroelastic response of high aspect ratio elastic wing excited by harmonic gust load. Paul et al. [23] designed a two-vane gust generator, measured gust velocity fluctuations using a hot-wire anemometer, and applied fast Fourier transform (FFT) to filter the experimental data, resulting in consistent values for gust frequency and velocity amplitude with theoretical predictions.

Although gusts can be generated in the flow field due to vanes oscillation, the flow directly behind the vanes is not stable. Generally, an even number of vanes are symmetrically arranged with the central axis of the wind tunnel, and the model to be tested is placed on the central axis. Another design of the gust generator is to use a rotary slotted cylinder mechanism, which is mainly composed of a slotted cylinder, diversion vanes and a drive motor. The slotted cylinder is used as a gust generator because of the gap, which affects the pressure difference between the front and back of the cylinder, thus affecting the resistance of the cylinder turbulence and the frequency of the vortex shedding. The flow field of the slotted cylinder the systematically measured by Igarashi [24], and an analysis was conducted to summarize its characteristics under various operational conditions. Olsen [25] focused on the influence of gap size and cylindrical contour shape on Strouhal number and resistance coefficient. Theoretical and experimental research was conducted by Tang et al. [26] on a rotary slotted cylinder gust generator, wherein the mechanism of gust generation and the relationship between gust and circular groove rotation frequency were analyzed. Wood et al. [27] used the NACA0015 airfoil to design a gust generator for a low-speed wind tunnel that can produce a gust at a frequency of 14 Hz. Brion et al. [28] conducted tests on a new gust generator device for wind tunnels, which can generate gust by oscillating airfoil, evaluating its performance at subsonic and transonic speeds. The device comprises a pair of airfoils oscillating up to 80 Hz for incoming flow velocities ranging from Mach 0.3 to 0.73. A proper evaluation of the gust generator was proposed by Zhao et al. [29] to develop experimental capability for full-scale aircraft model gust load alleviation tests.

This paper introduces design of a novel small-scale gust generator to generate Sears-type gusts, whose velocity fluctuation is dominantly perpendicular to the mean freestream flow [30]. The new gust generator has two vanes whose movement is driven by a cam. This form of gust generator has the advantages of simple structure, simple composition mechanism, convenient manufacture and assembly. Besides, the control system is simpler than conventional motor-driven generators. The specific movement of the vanes is controlled by the shape of the cam, which does not need sophisticated control systems for regulating the operation of the brushless motors, as was needed by previous gust generators. Thus, a common motor is adequate to provide the power to the gust generator. However, we found in the experiment that the system unavoidably vibrated at frequencies above 10 Hz, generating significant noise. Both the mechanical vibration and noise played an adverse role in generating the desired gusty flow field.

## 2. Gust Generator Facility Development

### 2.1. Design Scheme

A schematic sketch of the driving mechanism of the novel gust generator is shown in Figure 1. In order to comply with the prescribed motion law, the mechanism of cam rotation is employed. The pushing rod is connected to the pendulum rod at one end and to the cam follower at the other end, with the cam follower making contact with the working surface of the disc cam. The cam drives the reciprocating motion of the pushing rod, which in turn drives the pendulum rod and the cascade rotating along a fixed axis. The pitch motion of the cascade creates a sinusoidal gust that superposes the steady flow in the wind tunnel test section. To convert the oscillation into the linear motion, the contact between the pushing rod and pendulum rod is connected to a slider that moves along the pendulum rod. At a specific angular speed, the speed of the slider at the contact point between the pendulum rod and the pushing rod is constant, which is equal to the product of the angular speed $\omega$ and the distance $d$. Therefore, at a predetermined angular velocity, the pushing rod undergoes a rectilinear reciprocating motion at a constant velocity, ensuring the maximum pitch angle of the pendulum rod, $\theta_{max}$. Thus, the stroke of the pushing rod is $h = 2d \times \tan\theta_{max}$, where $d$ represents the perpendicular distance between the rotating center of the pendulum rod and the line where the reciprocating movement of the pushing rod is located. The geometric model of the gust generator was constructed in CATIA, as shown in Figure 2.

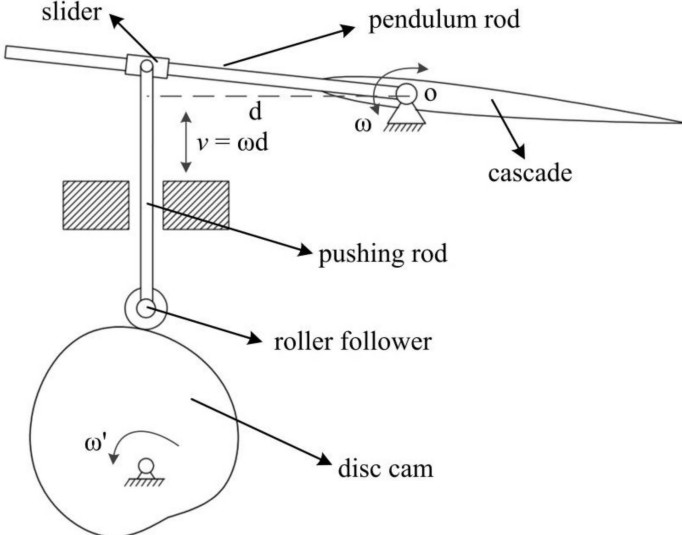

**Figure 1.** Working principle of the cam-based gust generator.

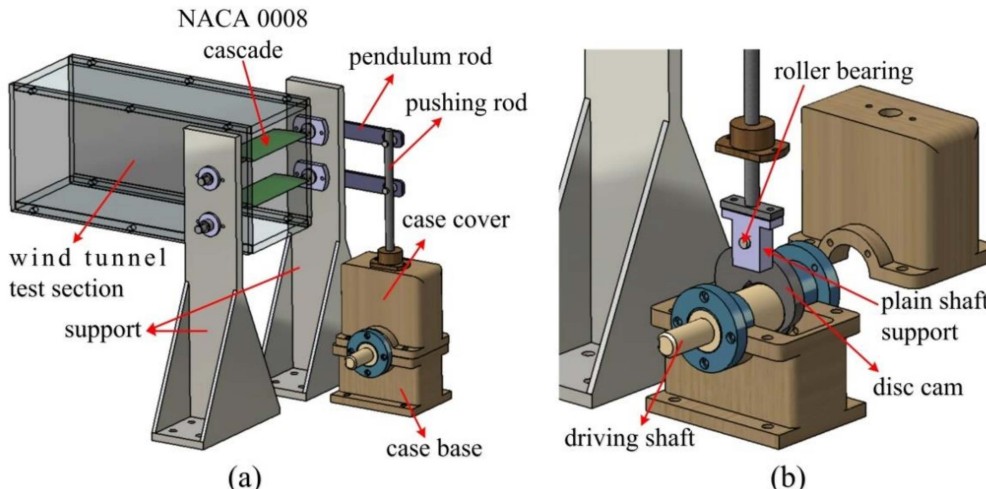

**Figure 2.** The geometric model of the gust generator: (**a**) assembly with the wind tunnel, (**b**) details of the driving device.

### 2.2. Wind Tunnel Facility

The gust experiment in this research was performed in the AF-1125 wind tunnel test stand of the Aircraft Engine Environmental Adaptation (AEEA) Laboratory at Nanjing University of Aeronautics and Astronautics (NUAA), as shown in Figure 3, which serves as a compact open wind tunnel specifically designed for the investigation of low-speed aerodynamics. The airflow enters the experimental section through a cellular rectifier with a maximum flow rate of 32 m/s. The experimental section has a dimension of 350 mm (length) × 145 mm (width) × 165 mm (height). The upper and lower wall panels of the experimental section consist of highly transparent glass with a thickness of 10 mm, facilitating the entry of the PIV laser into the field of view. The remaining two sides are made of highly transparent acrylic plates, providing convenience for observation and model installation.

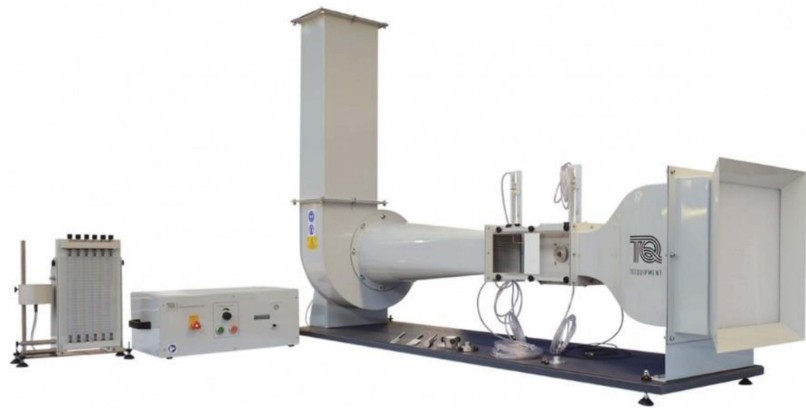

**Figure 3.** The AF-1125 low-speed wind tunnel of NUAA's AEEE group.

### 2.3. Gust Generator–Wind Tunnel Integration

A physical picture of the manufactured gust generator is shown in Figure 4a. Considering the limited room of the test section of the current wind tunnel, the relatively thin NACA 0008 airfoil was chosen as the profile of the two vanes of the gust generator. The whole unit is manufactured with the stainless steel material. The gust is positioned upstream of the wind tunnel test section, as shown in Figure 4b. The two vanes are symmetrical to the central plane of the wind tunnel. The support rods, which are situated 40 mm from the inlet of the test section, traverse through the holes on both sides of the acrylic plates within the wind tunnel.

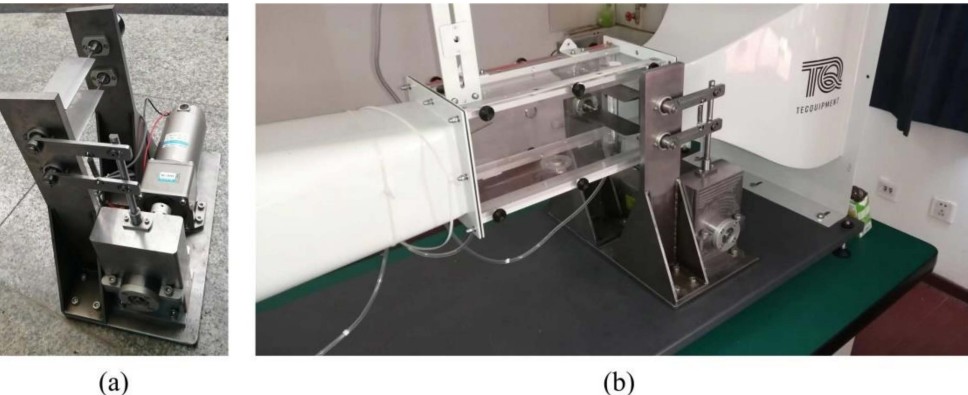

(a)                                                          (b)

**Figure 4.** Photos of the manufactured gust generator (**a**) and its integration with the test wind tunnel (**b**).

According to the dimensions of the wind tunnel test section that will be introduced next, it has been determined that the chord length of the gust generator vanes is 0.05 m, the span length is 0.14 m and the spacing is 0.06 m. The maximum pitch angle $A$ is set at $4°$, while ensuring that the vanes perform a uniform oscillation along the quarter chord length point.

*2.4. Experimental Setup*

PIV was employed to visualize the flow field generated by the gust generator, as shown in Figure 5. For the illumination of the flow field, a Nd:YAG laser was used, which can generate a dual-pulse laser at 532 nm, at a frequency up to 10,000 Hz. An articulated optical arm (laser guiding arm) transmitted the laser light to the regions of interest in the experiments. In addition, a sheet optic divergent is connected in front of the optical arm to convert the laser beam into an almost planar sheet laser, which was around 1.5 mm thick at its waist. The width of laser sheet is about 80 mm, which can cover the distance between the tail of the cascade and the center of the wind tunnel. A NAC ACS-3 high speed camera with 10,000 fps and a $1280 \times 864$ resolution was adapted with an 85 mm macro lens. The high-speed camera is synchronized with the laser through a synchronizer, enabling the camera to capture images at the same frequency as the laser. For the flow visualization studies, a thermally produced mineral oil fog was used as a flow tracer, which is called Di-Ethyl-Hexyl-Sebacat (DEHS). The average diameter of the tracer particles was only about 200 nm, which was small enough so that they had very low inertial tracking errors as they were convected in the vortical flow. A special nozzle was used to spray tracer particles from the entrance of the wind tunnel, and the experiment began after the tracer particles filled the test section.

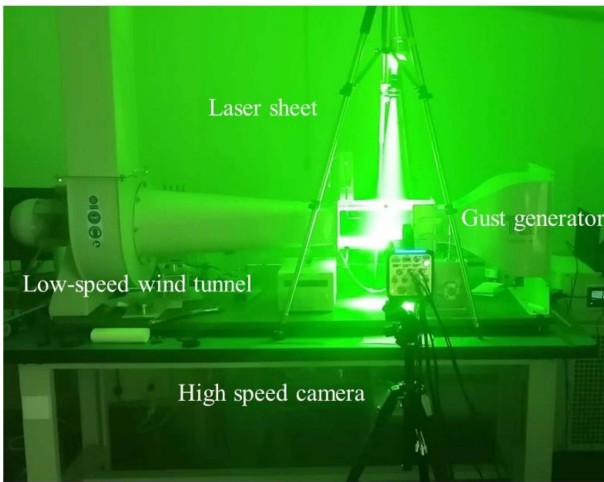

**Figure 5.** Setup picture of the measurement unit for gust experiment.

### 3. Numerical Simulation Methods

To help design the gust generator and validate the experimental results of its capability, two numerical simulation methods, OVM and FVM, were utilized to help determine the geometrical parameters of the oscillating vanes and the downstream test model for future gust response studies. This section introduces the two methods, whereas details of the methodological validations can be found in our previous work [31].

*3.1. Oscillating Vane Method (OVM)*

In order to simulate the gust field generated in a low-speed wind tunnel with a two-vane gust generator, determine the design parameters and optimize the design scheme of the gust generator, a two-dimensional flow field was established, as shown in Figure 6. The positive direction of the $x$ axis was consistent with the direction of the incoming flow, pointing downstream, and the $y$ axis was perpendicular to the flow direction. The height and length of flow field were the same as the experimental section of the wind tunnel, which were 0.145 m and 0.8 m, respectively. Two NACA0008 vanes symmetrical to the central plane of the wind tunnel were set in the upstream of the flow field. The chord length of the vanes was 0.05 m, and the spacing between the vanes was expressed as $D$. The flow field was partitioned into multi-block structured grids using the meshing tool Pointwise, and the entire computational domain was discretized into structured grids. The grid surrounding the vanes is presented in Figure 7. To accurately capture the flow characteristics within the viscous sublayer of the boundary layer, the height of the first layer of grid adjacent to the wall was set at $2 \times 10^{-5}$ m, meeting the requirement of standardized wall distance $y^+ < 1$. Additionally, each vane was encompassed by a perpendicular O-shaped sub-grid that grows at a rate of 1.1. To confirm the necessary resolution to achieve grid-independent results, a grid independence examination was conducted with five different grids. The amplitude of the gust velocity $\widetilde{v}$ generated under the condition of A = 4°, freestream velocity $U_\infty$ = 10 m/s and oscillation frequency $f$ = 10 Hz is compared and shown in Figure 8. As the number of grid cells reaches 60,000, $\widetilde{v}$ has shown a negligible difference with the results of finer girds. Considering both the accuracy and computational cost, the later simulations adopted the grid with 60,000 cells. The simulation was implemented using the angular Oscillating Displacement function in the open source computational fluid dynamics software package OpenFOAM v1912. The quarter chord of each vane was set as the center of rotation, causing the vanes to follow sinusoidal oscillations for generating Sears-type gusts [30]. The k-ω shear stress transport (SST) turbulence model was used, which has shown a wide adaptation to the adverse pressure gradient flow near the wall and flow separation state in a fully turbulent recirculating flow. The PimpleFoam solver, a finite-volume method-based solver in the open-source computational fluid dynamics software package OpenFOAM, was employed to solve the Navier–Stokes equation for the unsteady two-dimensional flow field. The coupling between pressure and velocity was computed using the PIMPLE (pressure-implicit method for pressure linked equations) algorithm, with no consideration given to other volume forces. In the finite volume discretization, the time term was discretized in a different format, to obtain flow field information at specific time points rather than average values within certain time steps. The time discretization method employed was the two-time step method, while the relaxation solution utilized a second-order backward Euler difference scheme. The discretization of the velocity terms, turbulence terms and energy terms was performed using a second-order linear upwind scheme. The pressure terms were discretized using a second-order linear difference scheme.

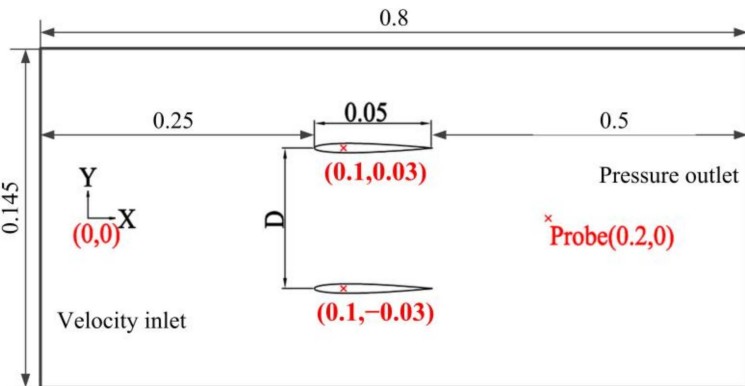

**Figure 6.** Sketch of the configuration and dimension of the flow field. The unit of length is meter.

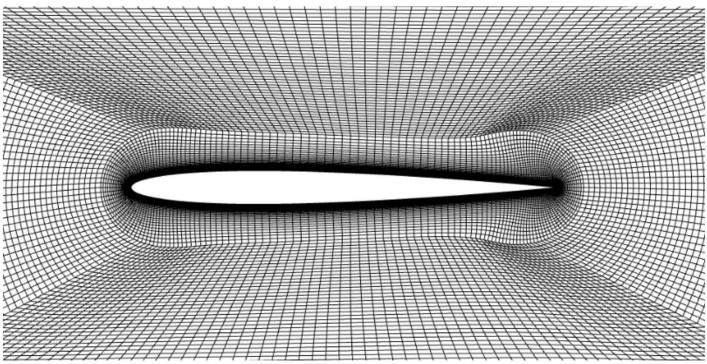

**Figure 7.** The grid surrounding each of the vanes.

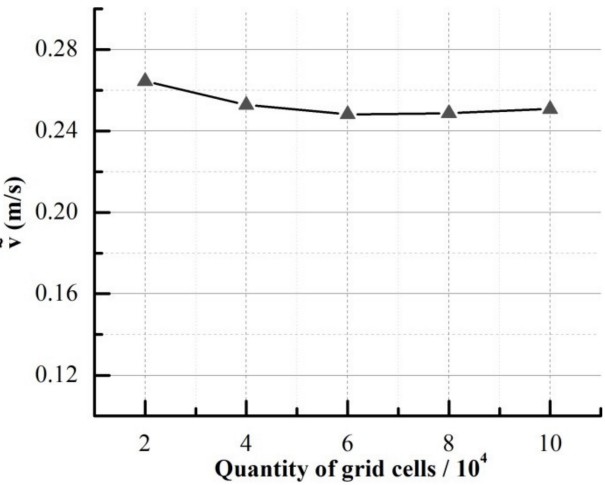

**Figure 8.** The results of the grid independence examination.

### 3.2. Field Velocity Method (FVM)

The field velocity method (FVM) is based on the arbitrary Lagrange–Euler (ALE) formula, which can capture unsteady flow conditions by calculating the pseudo-grid motion within the domain. It allows for the specifying of gust speeds by manipulating the grid time indicator without physically altering the grid. Mathematically speaking, considering the overall velocity, $\vec{v}$, in the computational domain, it can be reformulated as follows [7,31]:

$$\vec{v} = (u - x_\tau)i + (v - y_\tau)j + (w - z_\tau)k \tag{1}$$

The velocity components along the coordinate direction, respectively denoted by $u$, $v$ and $w$, are considered in conjunction with the grid time metric components, $x_\tau$, $y_\tau$ and $z_\tau$.

It should be noted that, for flows on a stationary body, these metric components are zero. The velocity field in the presence of horizontal gusts can be formulated as:

$$\vec{v} = (u - x_\tau)i + (v - y_\tau + v_g)j + (w - z_\tau)k \tag{2}$$

Therefore, the revised time measure is:

$$\widetilde{x}_\tau i + \widetilde{y}_\tau j + \widetilde{z}_\tau k = x_\tau i + (y_\tau - v_g)j + z_\tau k \tag{3}$$

Assuming that the horizontal sinusoidal gust in the study has the oscillation amplitude $\widetilde{v}$ and the frequency $f$, and the average velocity is $(u_0, v_0, w_0)$, the vertical component of gust inflow can be expressed as follows:

$$v = v_0 + \widetilde{v}_g \cdot sin(2\pi f \cdot t) \tag{4}$$

The time vector, denoted as $= [0 : T/N : (N-1)T/N]$, represents the discretization of gust velocity within a gust period $T$. The number of gust discretization intervals, represented by $N$, determines the resolution of the gust shape. A larger value for $N$ indicates a closer approximation to the physically continuous gust shape. In this study, the Sears sinusoidal gust is transformed into a superposition of a sinusoidal time-varying velocity field perpendicular to the steady freestream flow:

$$\vec{v} = U_\infty i + \widetilde{v}sin(2\pi f \cdot t)j \tag{5}$$

The velocity vector on the boundary is denoted as $\vec{v}$, while $U_\infty$ represents the magnitude of freestream flow velocity. Additionally, $\widetilde{v}$ and $f$ denote the amplitude and frequency of sinusoidal gusts, respectively. Figure 9a illustrates the gust velocity profile, which displays a single complete period of gust oscillation overlapped with a uniform incoming flow. To simulate the unsteady gusts, the gust velocity was discretized in python, on the basis of the steady-state flow field. During these simulations, the boundaries were modified to feed in gust velocities, as depicted in Figure 9b. Figure 10 shows the flowchart of the implementation of the FVM in OpenFOAM [31]. The first step involves conducting a steady state calculation of the basic flow field in the absence of gust, followed by the unsteady calculation with gust, where the specific gust velocity, generated by Matlab, is input at each time step.

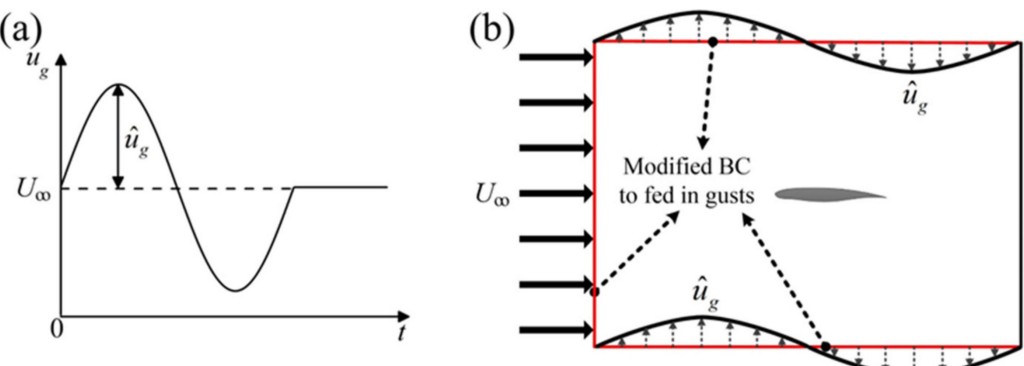

**Figure 9.** Sketch of (**a**) the gust velocity shape and (**b**) implementation of gust introduction in the FVM method [31].

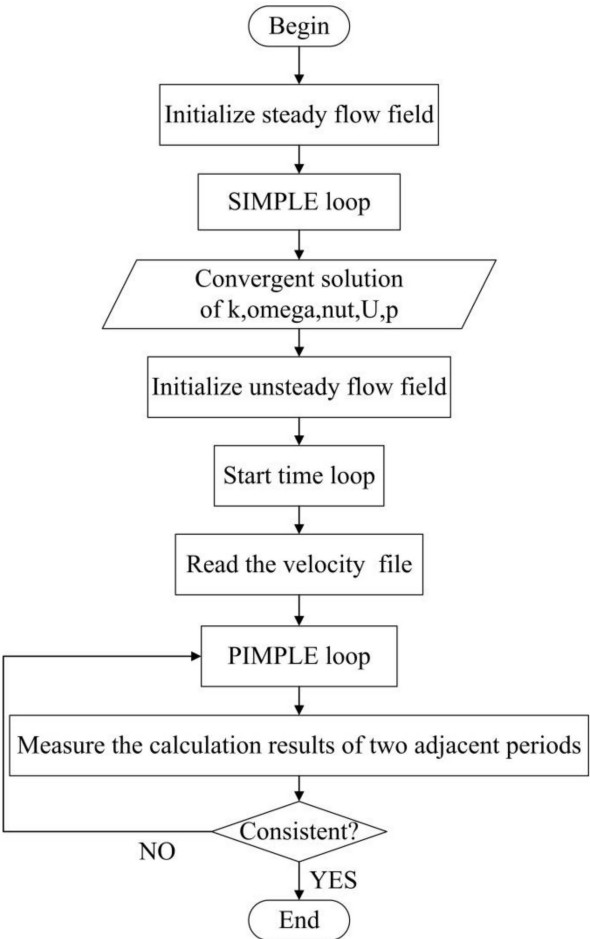

**Figure 10.** Flowchart of the FVM implementation in OpenFOAM.

## 4. Results and Discussion

### 4.1. Parametric Influences on Gust Generator Setup

#### 4.1.1. Impact of Vane Pitch Angle

A study conducted by Wu et al. [30] revealed that excessive pitch angles of a rotating vane can cause flow separation near the vane's surface, leading to the downstream transmission of vortices, affecting the wind gust shapes. The impact of flow separation decreases with an increasing vane oscillation frequency at the same flow velocity, while it increases with an increasing flow velocity at the same frequency.

Figure 11a–c, respectively, shows the numerical results of the pressure field and streamline distribution at maximum pitch angles of 4°, 6° and 8° under the conditions of a flow velocity, $U_\infty$, of 10 m/s and a vane oscillation frequency, $f$, of 10 Hz. The reference pressure is 101 kPa. It can be found that a significant flow separation occurs above the vanes at pitch angle of 6° and intensifies at 8°, which will be transmitted to the downstream and affect the gust. Figure 12 plots the gusty flow velocity at the oscillation frequencies of 4 Hz and 10 Hz while the pitch angle is fixed at 6°. It indicates the velocity fluctuation at the downstream monitoring point (0.2,0) for future gust response studies, thereby confirming the conclusion that the adverse influence of flow separation on the generated gust shape diminishes with increasing vane oscillation frequencies.

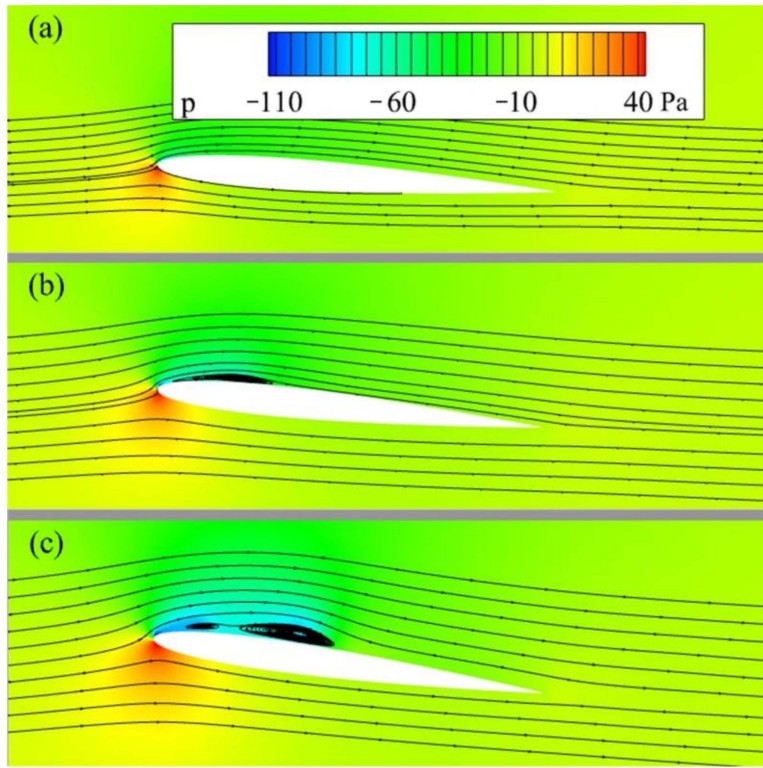

**Figure 11.** Flow separation occurring at the back of the vanes due to excessive pitch angles. (**a**–**c**) correspond to 4°, 6° and 8°, respectively.

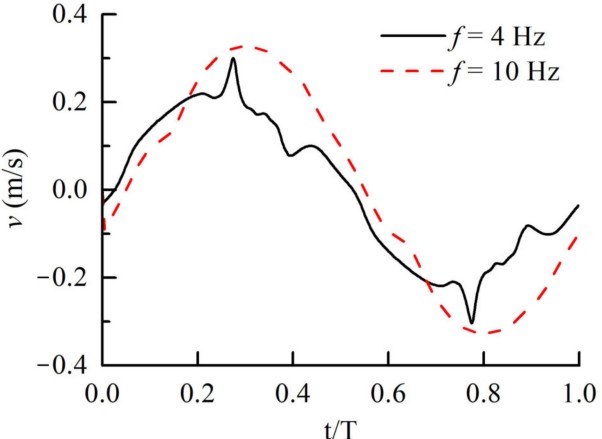

**Figure 12.** Comparison of gust velocity $v$ at the monitoring point for different oscillation frequencies and at $U_\infty$ = 10 m/s and pitch angle of 6°.

4.1.2. Impact of Vane Vertical Spacing

In order to reduce the interferences between the two vanes as well as between each vane and tunnel walls, a numerical simulation was carried out for the vertical spacings between the two vanes, $D$, of 30 mm, 40 mm, 50 mm and 60 mm and at the maximum vane pitch angle, $A$, of 4°, a flow velocity, $U_\infty$, of 10 m/s, and a vane oscillation frequency, $f$, of 8 Hz. The velocity $v$ in the $y$ direction at coordinates (0.2,0) was measured for three periods, as shown in Figure 13. With the increase in vane spacing $D$, the amplitude of the gust velocity in the $y$ direction, $v$, has little change. In comparison with the case of $D$ = 60 mm, the amplitude of gust velocity at $D$ = 30 mm only decreases by 3.8%. However, the shape of the gust velocity profile becomes smoother and closer to a sinusoidal gust pattern as the spacing between the two vanes increases, as shown in Figure 14.

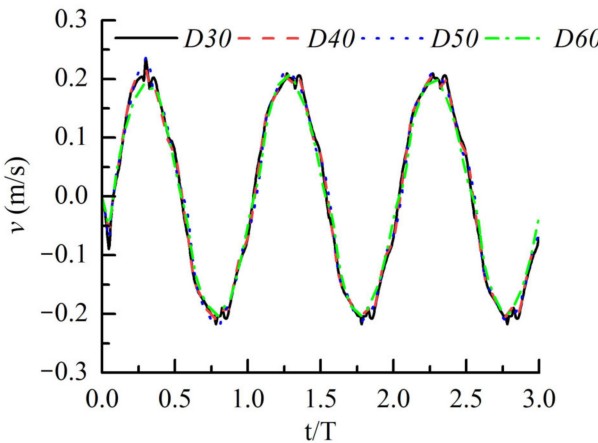

**Figure 13.** The effect of vane spacing on gust velocity magnitude.

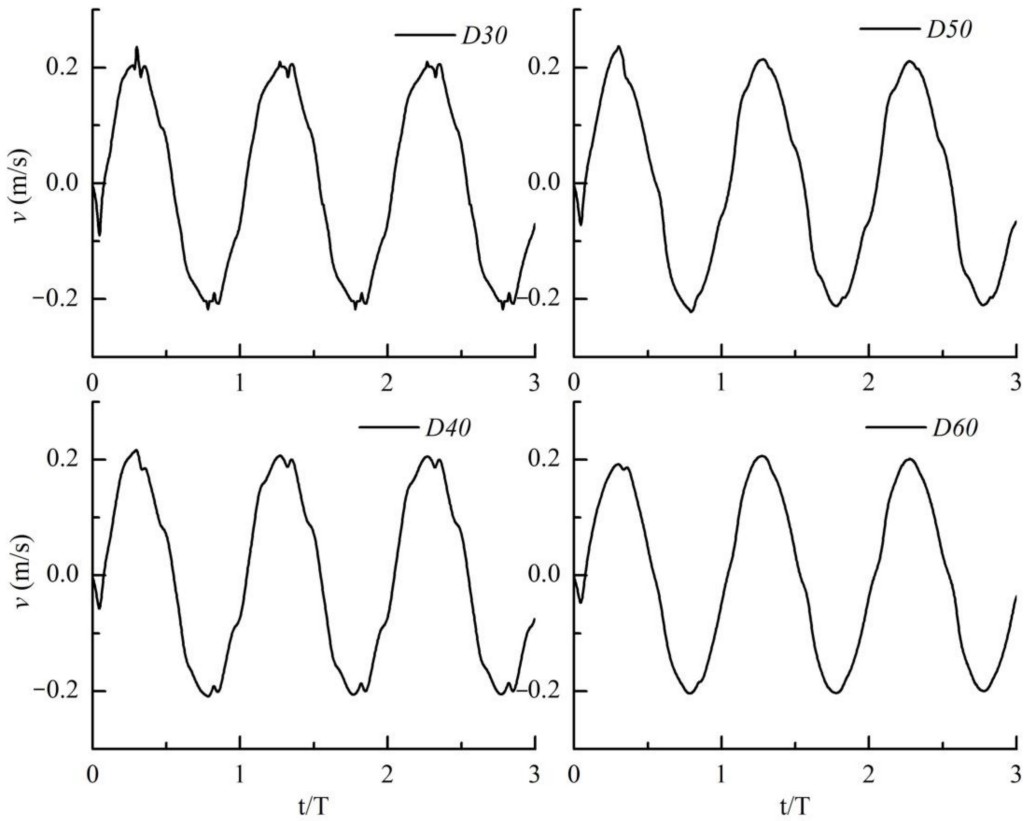

**Figure 14.** The effect of vane spacing on gust velocity shape.

### 4.1.3. Impact of Vane Oscillation Frequency

Numerical simulation was conducted under various oscillation frequencies, while the vane pitch angle $A$ was set to 4°, and the vane spacing $D$ set at 60 mm. Figures 15 and 16 illustrate the variations in gust velocity $v$ at the monitoring point, with respect to vane oscillation frequencies in three cycles for two flow velocities $U_\infty$. The amplitude $\tilde{v}$ increases with the rise in vane oscillation frequency at a fixed incoming flow velocity, resulting in a smoother gust curve that closely resembles a sinusoidal gust form. Figure 17 illustrates the velocity fluctuation at the monitoring point under different incoming flow velocities at a fixed oscillation frequency of 10 Hz. With an increasing velocity, the relative effect becomes smaller. Figure 18 plots the amplitude of gust velocity, $\tilde{v}$, corresponding to different combinations of incoming velocity and frequency. For a given incoming flow speed, the amplitude increases with higher frequencies. Similarly, it rises with the increasing in-

coming flow speed at a constant frequency. However, under low-speed and low-frequency conditions, the gust curve appears irregular due to the oscillation of the separated flow above of the vanes.

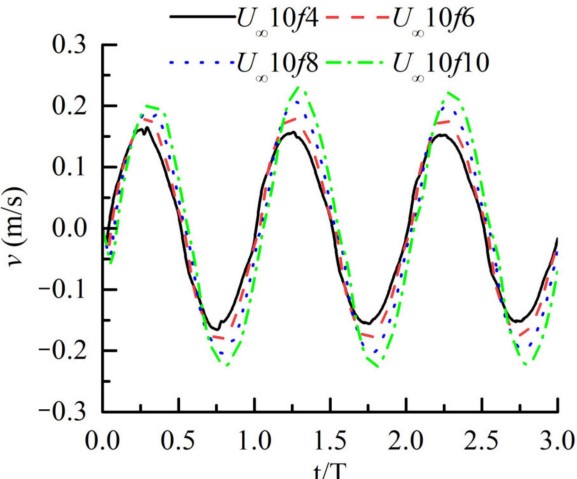

**Figure 15.** The effect of vane oscillation frequency on gust velocity at freestream velocity of 10 m/s.

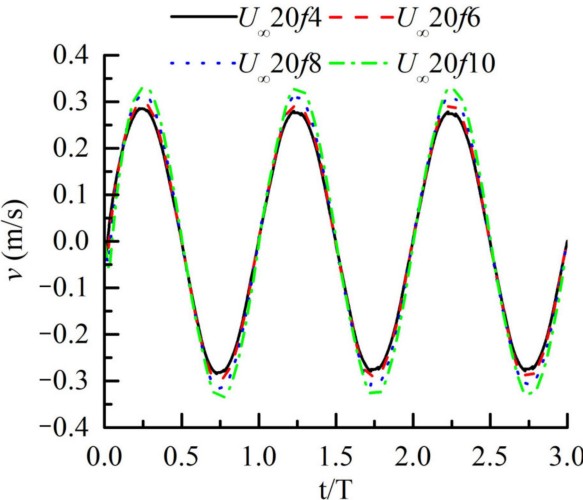

**Figure 16.** The effect of vane oscillation frequency on gust velocity at freestream velocity of 20 m/s.

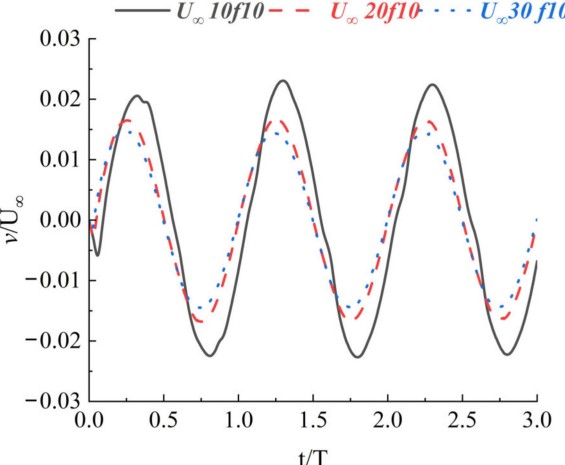

**Figure 17.** The effect of incoming flow velocity on gust velocity at oscillation frequency of 10 Hz.

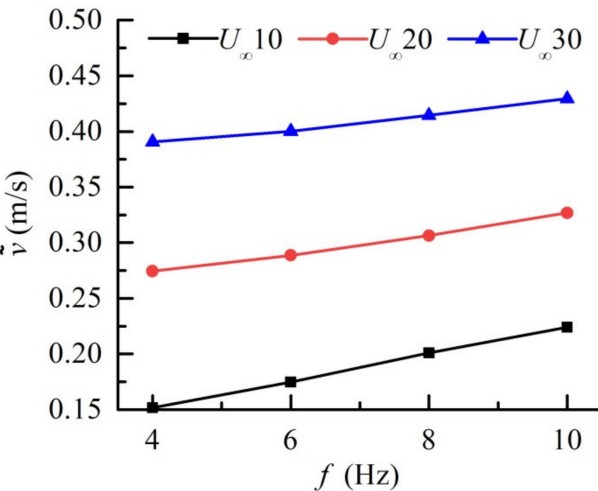

**Figure 18.** The relationship between the amplitude of gust velocity $\tilde{v}$, oscillation frequency $f$ and freestream velocity $U_\infty$.

### 4.1.4. Impact of Test Model Position

The oscillation of the vanes generated flow separation vortices on the back of the vanes, which convected downstream in the wake and affected the quality of the gusty flow field. According to the research of Wu et al. [31], in order to generate gusts consistent with the OVM method using the FVM, it is necessary to first calculate the amplitude of gusts generated by OVM, then set the parameter value $\alpha$ in FVM, remove the vanes from the OVM grid, regenerate the grid, and validate the gusts generated by FVM method. $\alpha$ represents the angle of gust which can be expressed as $\alpha = \arctan\left\{\frac{v}{U_\infty}\right\}$, where $v$ is the component of gust velocity in the $y$ direction, and $U_\infty$ is the incoming flow velocity. Figure 19 illustrates the utilization of FVM to measure the changes in the vertical gust velocity along the $x$ direction on the central axis of the flow field for different gust angles, $\alpha$. The incoming flow velocity, $U_\infty$, is 30 m/s and gust frequency, $f$, is 10 Hz. The horizontal coordinate $x$ represents the distance between the measured point and the trailing edge of the vanes. As both the upper and lower boundary conditions of the computed flow field are the tunnel walls, the gusts generated from the trailing edge of the vanes will dissipate in the wake flow and the gust velocity in the $y$ direction, $v$, will gradually decrease along $x$ direction. The setting of multiple $\alpha$ values for numerical simulation, extraction of monitoring points, and comparison with OVM is illustrated in Figure 20 to determine the corresponding $\alpha$ for the gust generated by OVM. In Figure 21, the fluctuation of velocity $v$ between FVM and OVM is presented over three cycles under the incoming flow velocity of 30 m/s and gust frequency of 10 Hz. It can be observed that both the velocity fluctuation amplitude and period at the monitoring point are consistent between FVM and OVM.

It is worth noting that the FVM approach, unlike the grid used in OVM for generating gusts, modifies boundary velocity conditions to generate gusts without the need for a dynamic vane grid that changes over time. As a result, there is no requirement for using a high-precision grid throughout the entire domain to transmit gusts from the inlet to the measurement area, leading to a significant reduction in computational effort. In this study, the total mesh count utilized in the vane oscillation method is approximately 60,000, with the mesh surrounding the vane dynamically changing over time steps. Conversely, the FVM employs around 8000 grids for calculation purposes and only requires modifications to boundary conditions, resulting in a significant reduction in computational time. On an equivalent computing platform, more than one hundred cores are required for the vane oscillation method, while less than one core suffices for the FVM.

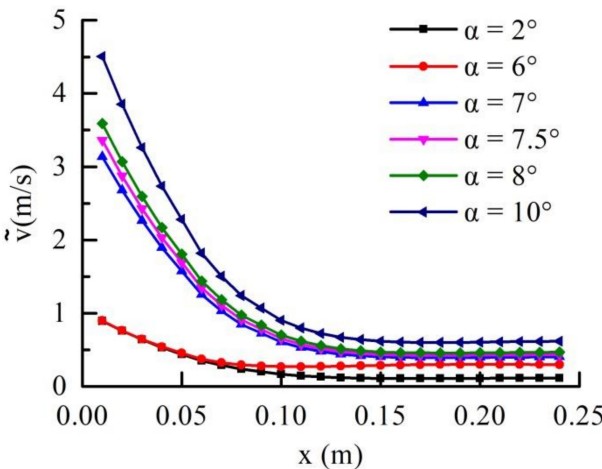

**Figure 19.** The amplitude of gust velocity, $\tilde{v}$, varies along the *x* direction in the FVM when different values of *α* are assigned.

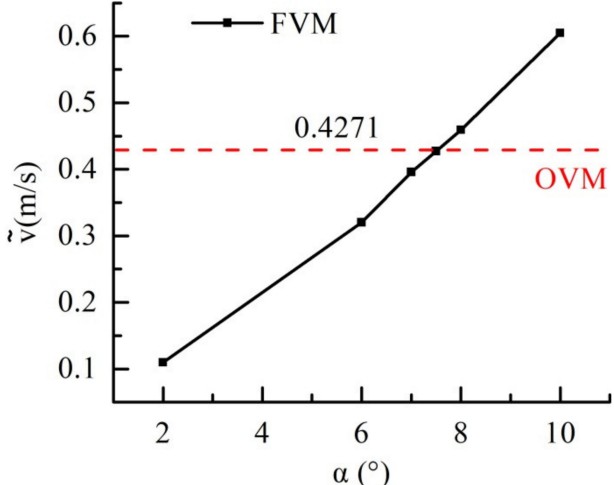

**Figure 20.** Comparison of gust velocity amplitude at monitoring points under different gust angles.

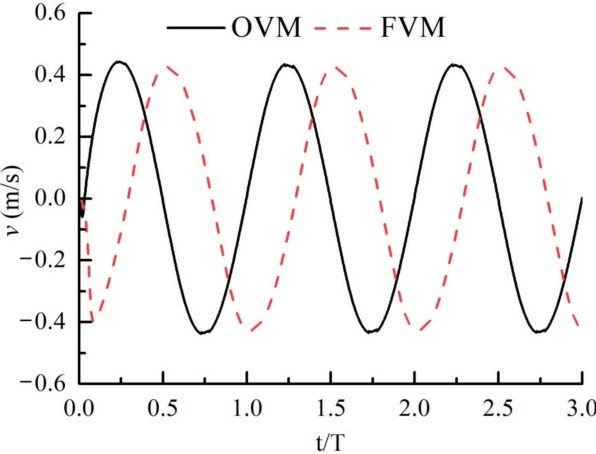

**Figure 21.** Comparison of gust effect produced by two methods.

### 4.2. Examine of the Capability of the Gust Generator

The fabricated gust generator was assembled with the wind tunnel to test its capability of simulating Sears-type gusts. The maximum oscillation amplitude of the vanes was fixed at 4°, while the oscillation frequency changed within 10 Hz. To accelerate the processing

of the flow field, two programs were developed to convert the coordinates and velocity files into the format that can be read by the postprocessing software, Tecplot, which can be downloaded from the Github website (https://github.com/Gravity-Zero-o/PIV_data_conversion, (accessed on 30 May 2023)). Figures 22–24 show the characteristic contours of the flow field generated by the gust generator in a certain instance. It can be obviously seen that the flow field at the presence of the oscillating vanes is no longer steady. Instead, the flow field becomes fully unsteady, due to the transport of a series of vortices generated by the gust generator downstream. The vortices shed from the trailing edge of the vanes caused significant fluctuations in the flow field, especially in the area right downstream of the vanes. It is worth noting that Figure 23 shows the velocity vector diagram when the frequency is 6 Hz and the inlet velocity is 10 m/s. As can be seen from Figure 15, the maximum velocity component in the $y$ direction is only about 0.2 m/s, which is much smaller than the velocity component in the horizontal direction. Therefore, the fluctuating signature of the velocity vector field is not obvious.

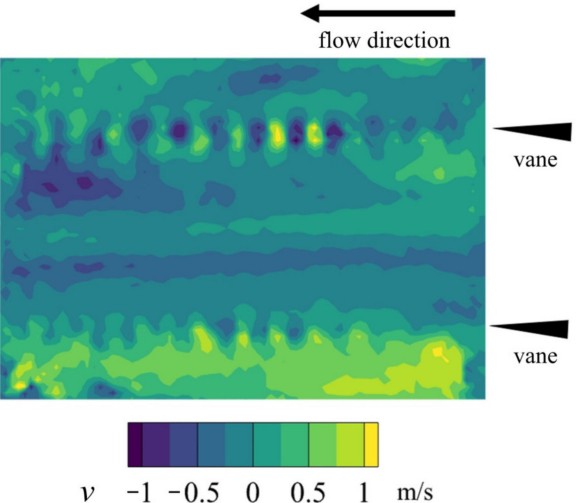

**Figure 22.** Velocity field generated by the gust generator at $f$ = 6 Hz and $U_\infty$ = 10 m/s.

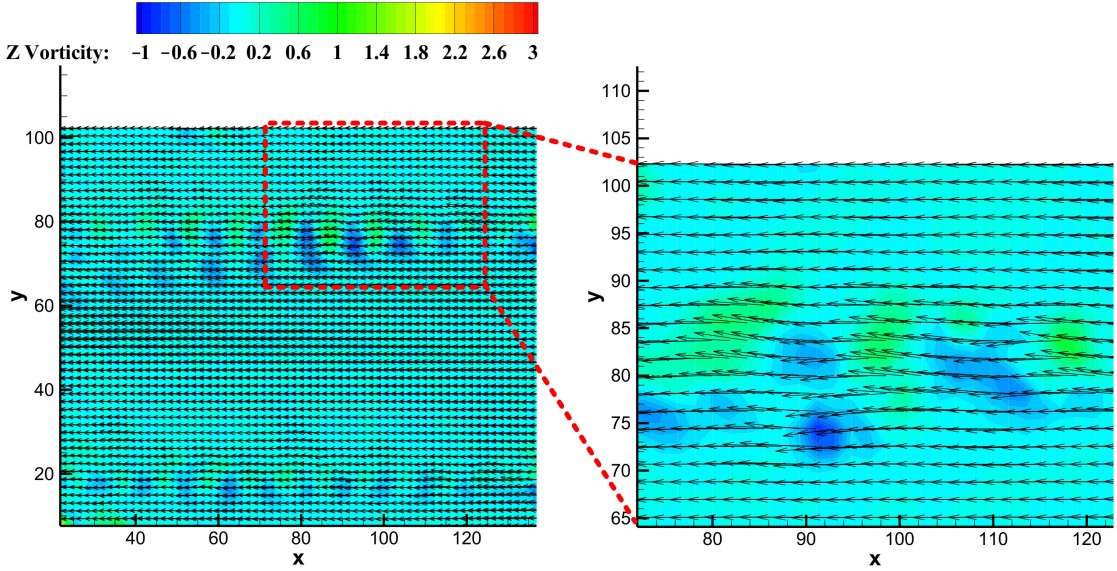

**Figure 23.** Vorticity field generated by the gust generator overlapped with the velocity vectors at $f$ = 6 Hz and $U_\infty$ = 10 m/s.

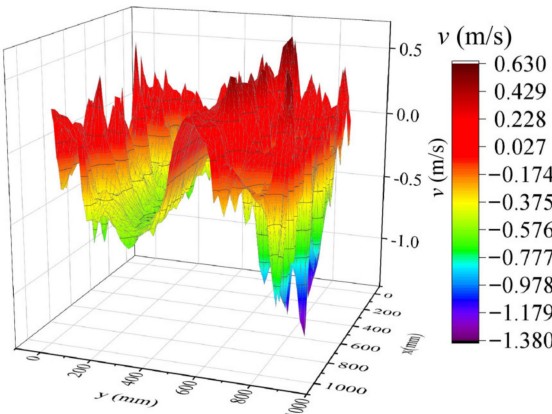

**Figure 24.** Iso surface of the velocity field generated by the gust generator at $f$ = 6 Hz and $U_\infty$ = 10 m/s.

It is because, in future gust response studies, the test model of interest will be positioned at the centerline of the flow field, that it is necessary to examine the status of the flow at the potential position of the test model. To this end, the flow velocity at three monitoring points, i.e., 60 mm, 70 mm and 80 mm downstream of the trailing edge of the vanes, were extracted from PIV. The responses of the vertical component of the flow velocity (i.e., the Sears gust velocity) to the gust generator at the same working condition as that in Figures 22–24 in 22 cycles are presented in Figure 25. Generally, it can be seen that the curves of the gust velocity at these three positions are highly overlapped with each other.

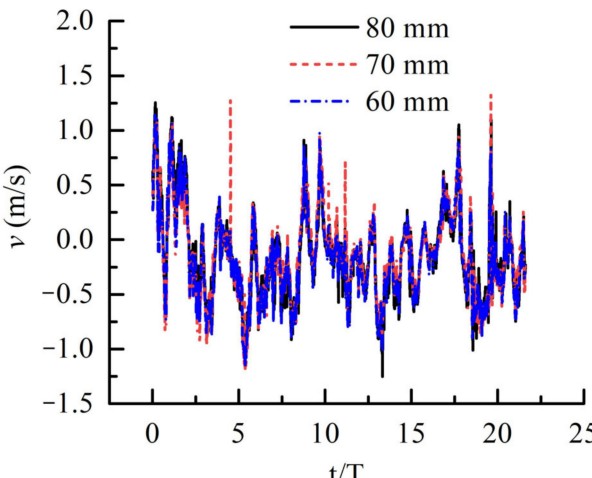

**Figure 25.** The gust velocity $v$ fluctuates in 21 cycles at three downstream positions of the vane.

The first 21 cycles of velocity were measured at a distance of 70 mm from the trailing edge of the vane. The gust generator produces vibrations during operation, which can interfere with the generated gust. To minimize this interference, we used a post-processing method described as follows: the twenty-one physical cycles are split into seven intervals, each having three physical cycles. To alleviate the interference signals, the seven signal intervals are superposed, to obtain an average value of the gust velocity results in the three physical cycles, as shown in Figure 26. The black curve represents the fluctuation of velocity $v$ at the measuring point over three cycles. The blue curve corresponds to the data filtered by using the fast Fourier transform (FFT) method [23], with the gust frequency set as the cutoff frequency for filtering the experimental data. On the other hand, the red curve illustrates the fluctuations in the velocity at the measuring point over three periods, obtained through numerical simulation using OVM. In general, the filtered experimental measurements agree well with the numerical results, especially in the later time when the flow field pattern becomes more regular. It is noted that the raw experimental data show

large fluctuations, which is considered to be due to the noise generated by the contact and impact between the components of the gust generator during operation. Thus, necessary measures are needed to prevent the mechanical noise from interfering the gusty flow field in future research. In conclusion, the novel gust generator developed in this study has acquired the expected capability of simulating Sears-type gusts.

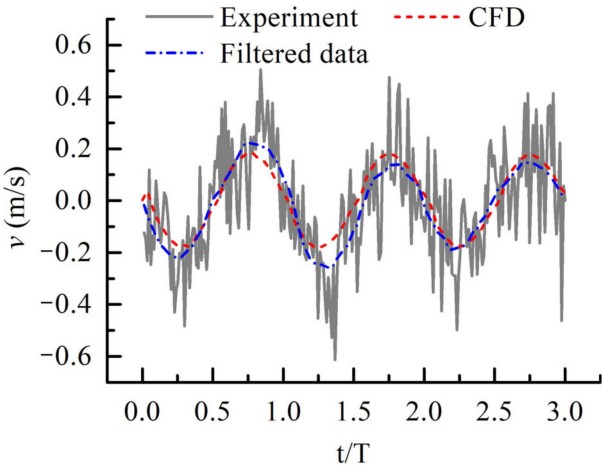

**Figure 26.** Comparison of measured gust velocity, filtering processing and numerical simulation results.

## 5. Conclusions

In this paper, a novel small-scale gust generator research facility is designed and examined for generating Sears-type gusts. Details about the design scheme, integration with the wind tunnel, experiment and validation of its capability with numerical methods have been presented. The following conclusions can be drawn:

(1) The effects of vane geometric and oscillation parameters on wind gusts points of vanes were simulated through numerical simulation. The influencing factors considered include the vane spacing $D$, maximum pitch angle $A$, oscillation frequency $f$ and incoming flow velocity $U_\infty$. The results obtained from the numerical simulations indicate that there is a positive correlation between the amplitude of gust velocity $\tilde{v}$ and vane pitch angle $A$, frequency $f$ and $U_\infty$, while there is a negative correlation with the spacing $D$. Nevertheless, excessively high velocity or large pitch angles can lead to flow separation, while insufficient spacing may result in the central gust of the flow field being influenced by the convecting vane wakes. Therefore, the amplitude of the gust can be adjusted by controlling the corresponding variables within a reasonable scope.

(2) The velocity boundary condition (FVM) was employed to generate gusts at the inlet of the computational domain. The variation in gusts with transmission distance is investigated. Through multiple tests, the amplitude of gust angle can be determined by FVM, ensuring consistency with the gust amplitude obtained using the OVM method.

(3) The PIV flow field measurement technique was employed to capture the actual operational performance of the gust generator under varying vane oscillation frequencies, $f$, and incoming flow velocity, $U_\infty$, during the wind tunnel tests. By comparing the experimental data with numerical simulation results, the capability of the currently developed gust generator is successfully validated. It should be noted, however, that necessary measures are needed to prevent the mechanical noise from interfering with the gusty flow field in future research.

**Author Contributions:** Conceptualization, Z.W. and H.T.; methodology, Z.W. and Y.G.; software, T.Z. and Y.G.; validation, Y.G.; formal analysis, Z.W. and T.Z.; investigation, T.Z. and Y.G.; resources, Z.W. and H.T.; data curation, T.Z. and Y.G.; writing—original draft preparation, Z.W. and T.Z.; writing—Z.W.; visualization, Y.G.; supervision, Z.W. and H.T.; project administration, Z.W. and H.T.;

funding acquisition, Z.W. and H.T. All authors have read and agreed to the published version of the manuscript.

**Funding:** This research was funded by [the National Natural Science Foundation of China (NSFC)] grant number [12172174], [the National Program for Young Talents] grant number [YQR23019] and [the start-up fund of Nanjing University of Aeronautics and Astronautics] grant number [YQR23006]. And the APC was funded by [the National Natural Science Foundation of China (NSFC)] grant number [12172174].

**Data Availability Statement:** The data presented in this study are available on request from the corresponding author.

**Conflicts of Interest:** The authors declare that they have no known competing financial interests or personal relationships that could have appeared to influence the work reported in this paper.

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
