# Peer review of "Development of a Novel Small-Scale Gust Generator Research Facility"

_aerospace, doi:10.3390/aerospace11010095_

Round 1

Reviewer 1 Report

Comments and Suggestions for Authors

The topic of the paper is generally interesting, however some issues have to be addressed before publication:

1.       What is the novelty of the new gust generator compared to previous gust generators with vanes?

2.       Chapter 2.1 must be revised because there are duplications.

3.       It should be described shortly why NACA008 was chosen.

4.       What was the y+ value exactly?

5.       Was the oscillatinglinearMotion used indeed? According to the openfoam wiki the origin cannot be defined for this function.

6.       It is unclear how the motion of the vanes was achieved during the simulations? Was the velocity field rotated or the vanes? In fig. 7 the vane is not in 0 angle of attack while in fig 6. the vanes are at 0 angle of attack.

7.       The description of the FVM method is not clear and neither is the connection between the OVM and FVM

8.       What results are shown in chapter 3.1.1? They seem like steady simulations, is this true?

9.       Which velocity component is shown in fig 10?

10.   I do not agree that the amplitudes in fig 11. decrease with the increase of vane spacing? Could the authors provide more information on this?

11.   I recommend using non-dimensionalized velocity for fig. 15.

12.   What does theta stand for? alpha is the angle of gust isn’t it?

13.   If there are vortices in the velocity field, then why are all velocity vectors horizontals in fig 21.?

Comments on the Quality of English Language

The level of English is acceptable, however English proofreading is recommended.

Author Response

Dear Reviewer 1:

Please find a point-by-point response to your comments in the attachment. Thanks a lot for your hard work on our paper.

Best wishes,

Zhenlong Wu

Reviewer 2 Report

Comments and Suggestions for Authors

Review of
Development of a novel small-scale gust generator research facility
by
Zhenlong Wu, Tianyu Zhang, Yuan Gao and Huijun Tan

This manuscript describes numerical simulations and wind tunnel measurements for gust generators. The authors use 2D CFD simulations two perform a parameter study of a gust generator consisting of two oscillating vanes. They resolve the vane motion for this study. Based on these results, the authors also perform tests using the Field Velocity Method. An experimental study is also performed using a newly devised mechanism to produce vane oscillations.

Overall, the manuscript might be suitable as a conference proceedings paper, but it fails to meet the standards of a proper journal publication. The main issues are as follows:

- A grid dependency study is missing. The overall grid number of 60.000 seems quite low to properly resolve the flow around two moving airfoils, and to also resolve the unsteady flow field downstream.
- The authors fail to explain, why the newly devised experimental mechanism is even beneficial. What are disadvantages of producing the vane movement directly via brushless motors, as described e.g. in the work by Ricci and Scotti (ref. 8), or by a linear motor, as described in the paper by Liu et al. (ref. 9). It seems that the current mechanism leads to significant disadvantages, especially when comparing the amplitudes of the high frequency distortions with those of Ricci and Scotti.
- A proper study would include more than one test case for the new gust generator. The best scenario would be a parameter study where the oscillation frequency and the freestream velocity are varied. A comparison with already published data of vertical speeds, e.g. those of Ricci and Scotti, would be reasonable. With the current data the authors fail to demonstrate that the mechanism works properly. The authors claim that mechanical noise is mainly responsible for the strong high frequency distortions, but without demonstrating that (remark: would mechanical noise not be a consequence of vibrations of the mechanism itself ?).

There are also some further issues with the manuscript:

- The introduction section needs to be improved. A short description of the principles of the Field Velocity Method and the Oscillating Vane Method should be provided. The four studies cited after the terms OVM and FVM are introduced only cover the FVM and the Velocity Splitting Method (which was not mentioned before at all).  Only later, two papers are cited, which apply the OVM. In a very short literature survey other relevant studies using the OVM can be found (e.g. French et al., 2021, doi: 10.3390/wind1010004, Yigili et al., 2022, doi:10.1088/1742-6596/2265/2/022108). A relevant experimental study which includes a proper evaluation of the gust generator was published by Zhao et al., 2022, doi: 10.1016/j.cja.2022.06.016

- The description of the mechanism depicted in Figs. 1 and 2 should use the same terms as those used in the figures. Is it reasonable to use the same symbol ‘omega’ for two different rotation rates in Fig 1? Why is the disc not circular or elliptic, but contains a bump?

- The dimensions of the computational domain are not provided. The values of the y+ values are not given.

- Line 199f: ‘The k-omega shear stress transport (SST) turbulence model was used, which can reflect the wall stall and flow separation state in a fully turbulent recirculating flow.’ What is ‘wall stall’, and why can it be reflected?

- The nomenclature and the wording of the FVM description are actually based on the work by Singh and Baeder, 1997, doi: 10.2514/2.2214, which would deserve a citation. The unit vectors would need to be identified as vectors. Why are horizontal sinusoidal gusts introduced in eq. (2). You use vertical gusts in your study.

- What is the purpose of Figure 12? Figure 11 already demonstrates that the vane distance does hardly influence the velocity field at the evaluation point.

- What is the frequency of the vortex streets shown in Figs. 20 and 21? Can you find that frequency in the data shown in Fig. 24? If so, it might not only be noise (or friction effects of the slider?) that leads to the high levels of high frequency fluctuations observed.

Comments on the Quality of English Language

Wording and grammar require proper language editing. The following is a list of grammatical errors, faulty expressions, just up to line 30 (the language quality of the remainder of the manuscript is not better).

-          Line 8ff: A novel small-scale gust generator research facility was developed and examined for generating Sears-type gusts. (note: the facility was not developed ‘in this paper’) The design scheme, its integration with a wind tunnel, the conducted experiments and the validation of its capabilities are presented in detail in this paper.

- Line 13ff:  … were both utilized to investigate the impact of geometrical parameters of pairs of oscillating vanes on the velocity field downstream. (note: the were not jointly used, jointly would imply that the methods were used together at the same time). Filtered experimental measurement results filtered agree well with the numerical data, validating the capability of the developed gust generator to produce sinusoidal gusts.

-          Line 16f: However, it should be noted that further measures are needed to prevent mechanical noise from interfering with the gusty flow field, which will be a focus of future research.

-          Line 20: a major factor in numerous catastrophic flight accidents

-          Line 26f: numerical approaches based on computational fluid dynamics (CFD) simulations (note: ‘numerical approaches’ and ‘computational’ were already mentioned)

-          Line 27f: yielding numerous significant results

-          Line 30ff: Li et al. [3] applied the field velocity method in unsteady Reynolds averaged Navier-Stokes (URANS) simulations to predict the responses of an airfoil to arbitrarily shaped gust penetrations. (note: the sentence was nearly a 1:1 copy of the sentence of the cited paper, you should not do that)

Author Response

Dear Reviewer 2:

Please find a point-by-point response to your comments in the attachment. Thanks a lot for your hard work on our paper.

Best wishes,

Zhenlong Wu

Reviewer 3 Report

Comments and Suggestions for Authors

Present paper shows the numerical simulation of the airfoil in pitching motion.  Two vanes are set  in order to generate the gust velocity components. The effect of the relative distance between the vanes are investigated. The velocity and vorticity field after the vanes are also investigated using the PIV measurements. Present study shows good results by using both experimental and numerical methods. I'd like to ask several questions and give comments for the present manuscript.

1. I am wondering what is the definition of the swing angle. Is it different from the pitch angle?

2. The streamline pattern after the TE of the vane in the Fig. 9 look like steady ones. As shown in the Fig.20, the streamline should show the wavy patterns. Is it because the small amplitude of the transverse velocity component ?

3.  In the figure, the gust velocity is less fluctuating for smaller frequency. Conventionally high frequency motion can produce the more fluctuating compontents. I am wondering if the authors can show the reason for that by showing the unsteady pressure and velocity distributions.

4. On the figure 11, the increase in the distance(vane spacing) smoothed out the velocity fluctuations. I think it is because the measurement point is locatted inbetween the vanes. What if the measurement point is located at the same height as the pitching axis.  

5. The numerical methods in the present study did not validated. Please show the validation results and grid independence.

6. In the FVM, the whole flow fields  around the vanes are affected whereas, in the OVM, the flow fields around the vane are affected by the movement of the vanes. Thus, the domain size will be very important for FVM. 

7. Present vanes are in pitch oscillation. However, the authors used the oscillatingLinearMotion function. Why the authors did not use the oscillating RotatingMotion function in the OpenFoam? I am wondering if the authors can explain the oscillatingLinearMotion function.

8. On the line 406, the flow field is still unsteady because it is time varying. Its pattern becomes more regular.

9. Why velocity amplitudes in Fig. 23 and 24 are different from each other?

10. The measurement point is at a distance of 70mm from the TE of the vane which are positioned almost right after the vane. The motion of the vane can affect the aerodynamics of the other test models(wings, airplanes, etc.). Inspecting the results in the Fig. 17, the effect of the  gust velocity are almost smoothed out when the test model are at far downstream behind the vane. I am wondering what is the pros and cons of the present test facility, and the effective test range. 

Author Response

Dear Reviewer 3:

Please find a point-by-point response to your comments in the attachment. Thanks a lot for your hard work on our paper.

Best wishes,

Zhenlong Wu

Reviewer 4 Report

Comments and Suggestions for Authors

Dear authors, please find my comments in the attached file.

Author Response

There is no comment from Reviewer 4., who has recommended publication in its present form.

Round 2

Reviewer 1 Report

Comments and Suggestions for Authors

Improvements have been made, however there are still issues to be addressed:

1. I still do not understand why the right branch of Fig. 10 is needed? Why isn’t the OpenFOAM simulation enough? How and why is the speed generated in Matlab?

2. With the non-dimensionalized velocities, the description of fig 17. should be changed. With increasing the velocity the relative effect becomes smaller. (Of course in absolute values if you increase the velocity the velocity will be increased….that is not a scientific finding)

3. If the authors wish to demonstrate the vorticity field in fig 23. I recommend showing a smaller area, with higher density of vectors, so that the vortices can really be seen.

Comments on the Quality of English Language

English is OK.

Author Response

  1. I still do not understand why the right branch of Fig. 10 is needed? Why isn’t the OpenFOAM simulation enough? How and why is the speed generated in Matlab?

Answer: We are sorry that there is indeed some confusion here. Actually, the gust velocity file is generated by Matlab before gust calculation. During gust calculation, the value of the gust velocity is read by OpenFOAM at each time step. Thus, it is indeed unnecessary to have this branch in the flowchart.

  1. With the non-dimensionalized velocities, the description of fig 17. should be changed. With increasing the velocity the relative effect becomes smaller. (Of course in absolute values if you increase the velocity the velocity will be increased….that is not a scientific finding)

Answer: The sentence “With increasing the velocity the relative effect becomes smaller” has now replaced the original description of Fig. 17. Thanks.

  1. If the authors wish to demonstrate the vorticity field in fig 23. I recommend showing a smaller area, with higher density of vectors, so that the vortices can really be seen.

Answer: A close view of the local smaller area is added as a second sub-figure of Fig. 23. Thanks for your suggestion.

Reviewer 3 Report

Comments and Suggestions for Authors

Present study is focusing on the generation of lateral gust. The manuscript is faithfully  revised as suggested by the reviewer.

Author Response

There is no revision comment from this reviewer.